# Effect of ECAP on Microstructure, Mechanical Properties, Corrosion Behavior, and Biocompatibility of Mg-Ca Alloy Composite

Song-Jeng Huang [1], Chih-Feng Wang [1], Murugan Subramani [1,*] and Fang-Yu Fan [2,*]

1   Department of Mechanical Engineering, National Taiwan University of Science and Technology, Taipei 106, Taiwan; sgjghuang@mail.ntust.edu.tw (S.-J.H.); d11003006@mail.ntust.edu.tw (C.-F.W.)
2   School of Dental Technology, College of Oral Medicine, Taipei Medical University, Taipei 101, Taiwan
*   Correspondence: smurugan2594@gmail.com (M.S.); fish884027@tmu.edu.tw (F.-Y.F.); Tel.: +886-905349551 (M.S.); +886-02-27361661 (ext. 5212) (F.-Y.F.)

**Abstract:** This study investigates the effects of incorporating MgO into magnesium–calcium (Mg-Ca) alloy composites and subjecting them to the equal channel angular pressing (ECAP) process on the resulting mechanical and corrosive properties, as well as biocompatibility. Initially, the incorporation of MgO into the Mg-Ca alloy composites did not yield significant improvements in grain refinement, tensile strength, or corrosion rate reduction, despite exhibiting improved biocompatibility. However, upon subjecting the Mg-Ca-MgO alloy composites to the ECAP process, noteworthy outcomes were observed. The ECAP process resulted in substantial grain refinement, leading to significant improvements in tensile strength. Furthermore, a marked decrease in corrosion rate was observed, indicating enhanced corrosion resistance. Additionally, the biocompatibility of the Mg-Ca-MgO alloy composites improved after undergoing the ECAP process. These findings highlight the synergistic effect of incorporating MgO and employing the ECAP process, providing valuable insights into the development of advanced magnesium-based materials with superior mechanical properties, reduced corrosion rates, and improved biocompatibility.

**Keywords:** magnesium–calcium alloy; MgO reinforcing phases; equal channel angular pressing; grain refinement; biocompatibility

## 1. Introduction

Bone defects can be repaired using various techniques, such as tissue engineering, gene therapy, physiotherapy, and other complementary methods, which include three essential components: cells, scaffolds, and growth factors. The design of material engineering focuses on the ability to replicate and meet the requirements of the host while achieving specific biomechanical properties. These materials are highly bioactive, easily integrated with the human body, and exhibit non-toxicity, non-allergenicity, and resistance to inflammation or foreign body reactions. Artificial bone replacement materials encompass metals, ceramics, natural polymers, artificial polymers, and blends, each offering advantages in terms of mechanical properties, biocompatibility, biodegradability, cell adhesion, and proliferation based on their unique characteristics.

Magnesium (Mg) and its alloys are highly valued in the automotive, electronics, and aerospace industries due to their low density, low elastic modulus, high strength-to-weight ratio, and fracture toughness [1–5]. Additionally, magnesium plays a significant role in various enzymatic reactions within the human body, participates in energy metabolism, maintains nerve conduction and muscle contraction, balances electrolytes, and supports physiological functions. Moreover, after degradation in the biological body, magnesium exhibits positive effects on bone cells, making it highly promising for the development of artificial bone restorations [6,7]. The consideration of corrosion resistance in magnesium

alloys significantly impacts their applicability in artificial restorations, leading to extensive research focusing on the development of Mg-Al and Mg-RE (rare-earth) systems [8,9]. According to the binary phase diagram of Mg-Ca, the maximum solubility of calcium in magnesium at 789.5 K is 1.34 wt%. Calcium (Ca), which is the main component of human bone, plays a crucial role in chemical signal transmission and possesses the advantage of low density (1.55 g/cm$^3$).

Z. Li et al. hypothesized that the simultaneous release of magnesium and calcium ions could have a positive impact on bone healing. They subjected the Mg-1Ca alloy to mechanical processing through rolling and extrusion, and then investigated its effects on mechanical properties and corrosion resistance. Excessive calcium content can significantly diminish the mechanical properties and corrosion resistance of magnesium alloys. Based on the findings, the optimal mechanical properties and corrosion resistance (12.98 mm/yr) were achieved when the calcium content was 1% wt. [10]. Moreover, the addition of reinforcing phases to magnesium alloys in the form of metal matrix composites (MMCs) offers benefits such as cost-effectiveness, improved mechanical properties, and corrosion resistance for industrial applications. Biocompatible phases, such as hydroxyapatite (HAp), tricalcium phosphate ($\beta$-TCP), silica oxide (SiO$_2$), and magnesium oxide (MgO), which are components of bioglass, are commonly discussed in this context [11,12].

Furthermore, the mixture of metal oxides (MMO) can serve as a solid catalyst for glycerol etherification. Talebian-Kiakalaieh et al. conducted a study where they demonstrated that the presence of acidity and alkalinity enhances the molecular interaction and activation of glycerol on the catalyst surface [13]. The high acidity of mixed oxides can facilitate glycerol conversion through dehydration, while alkaline earth elements play a role as well. The correlation between catalytic activity and alkalinity was observed in metal oxides, with the glycerol conversion rate increasing in the following order: MgO < CaO < SrO < BaO [14]. The combination of calcium with MgO particles resulted in grain refinement of Mg alloys [15]. Additionally, the degradation of magnesium oxide in the human body promotes bone cell growth and is beneficial for bone tissue regeneration [16,17]. Therefore, in this study, we will attempt to incorporate MgO as a reinforcement material.

Stir casting, also known as the stir casting process or mechanical stirring, is a widely employed manufacturing technique used for producing MMCs [18,19]. This technique involves mechanically stirring reinforcing particles, such as ceramic or metallic particles, into a molten metal matrix. Additionally, mechanical processing often utilizes severe plastic deformation (SPD) to enhance mechanical properties by inducing significant strains during the deformation process. Among the different deformation methods, equal channel angular pressing (ECAP) is a commonly used and convenient approach [20,21]. In their study, Hosaka et al. prepared ECAP specimens by cutting the AZ31 ingot and subjecting it to annealing treatment at 723 K for 24 h. Subsequently, they conducted immersion tests on the specimens processed at two different ECAP temperatures (573 K and 423 K) after undergoing eight passes of pressing. RPMI-1640 solution at 310 K was used as a simulated corrosion environment in the human body. Experimental results revealed that the sample processed at 573 K exhibited dynamic recrystallization, resulting in larger grain size compared to the sample processed at 423 K. Moreover, the processing temperature of 423 K induced greater dislocations in the material during the pressing process, leading to increased residual stress and hardness [22]. This process can stimulate dynamic grain recrystallization, thereby refining the grain structure and enhancing mechanical properties and corrosion resistance [23,24].

Based on the above discussion, the objective of this study is to offer a thorough comprehension of the Mg-Ca-MgO composite by investigating its synthesis techniques, as well as its mechanical, corrosion, and biomedical properties, considering the influence of the ECAP process.

## 2. Materials and Methods

The magnesium–calcium (Mg-Ca) alloy, obtained from Suzhou Haichuan Rare Metal Products Co., Ltd., Jiangsu, China was used as the matrix material with varying calcium content ratios (Ca = 0.7%, 0.9%, 1.1%). To reinforce the alloy, 2% of MgO particles with a particle size of 30 nm were incorporated and purchased from Diyi Chemical Materials Co., Ltd., Taipei, Taiwan. The composites were fabricated using the stir-casting method, which was previously described in our research [18]. After fabrication, the ingots were cut into dimensions of $10 \times 10 \times 5$ mm$^3$. To prepare the samples for microstructural analysis, they were ground using silicon carbide abrasive papers (#240, #400, #600, #1000, #2000), followed by polishing with aluminum oxide paper. Microstructural analysis was performed using an optical microscope (OM-Zeiss Axiotech 25HD, Jena, Germany), scanning electron microscope (SEM-JSM-7900F, Tokyo, Japan), and X-ray diffraction (XRD-Rigaku XtaLAB Synergy DW, Tokyo, Japan). For ECAP analyses, the ingots were cut into dimensions of 11.5 mm $\times$ 11.5 mm $\times$ 75 mm. Heat treatment was conducted in a furnace at 410 °C for 24 h. Following the heat treatment, ECAP processing was performed with the following parameters: channel angle $\Phi = 120°$, outer rounding angle $\Psi = 60°$, working temperature of 350 °C, Bc route, and extrusion after 4 passes.

The tensile test, which included measurements of yield tensile strength (YTS), ultimate tensile strength (UTS), and elongation, was conducted using an MTS 810 universal testing machine with a capacity of 100 kilonewtons. The tensile specimens were prepared in accordance with the ASTM E8 standard.

Corrosion analysis was carried out using a dynamic potential polarization curve test (using the PalmSens4C electrochemical analyzer/impedance analyzer). In this experiment, a three-electrode system was employed for measurement. The sample served as the working electrode, the platinum sheet was used as the counter electrode, and the saturated silver chloride electrode (Ag/AgCl sat. KCl) served as the reference electrode. Hank's Balanced Salt Solution (HBSS) was used as the solution, and its composition is provided in Table 1. The exposed area of the sample was 1 cm$^2$, with a sample area to solution volume ratio of 1:250. The solution temperature was maintained at $37 \pm 0.5$ °C. Prior to measurement, it was necessary to wait for the open circuit potential (OCP) between the sample and the electrolyte to reach dynamic equilibrium, which typically took approximately 15 min. Subsequently, the dynamic potential measurement was performed and the data were recorded. The experimental scan voltage range was set at $\pm 1$ V, with a scanning rate of 1 mV/s.

**Table 1.** The chemical composition of Hank's Balanced Salt Solution [25].

| Chemicals | g/L |
|-----------|-----|
| NaCl | 8.0 |
| KCl | 0.4 |
| $CaCl_2$ | 0.14 |
| $MgSO_4 \cdot 7H_2O$ | 0.06 |
| $MgCl_2 \cdot 6H_2O$ | 0.1 |
| $Na_2HPO_4 \cdot 12H_2O$ | 0.06 |
| $KH_2PO_4$ | 0.06 |
| $C_6H_{12}O_6$ | 1.0 |
| $NaHCO_3$ | 0.35 |

The cytotoxicity test and cell apoptosis test were used to investigate the biocompatibility analysis. The MC3TE-E1 cell line was cultured in Eagle Minimum Essential Medium (MEM, Gibco, Waltham, MA, USA) with 10% fetal bovine serum (FBS) solution at 37 °C. The proliferation and apoptosis of the cells were analyzed with the Cell Counting Kit-8 (CCK-8; Sigma, Taipei, Taiwan) and lactate dehydrogenase (LGH) assay.

## 3. Results and Discussion

### 3.1. Materials Characterization

Microstructural Analysis

Figure 1a illustrates the optical microstructure (OM) of Mg-1.1%Ca alloy composites. It discloses the presence of $Mg_2Ca$ intermetallic phases at the grain boundary (indicated by the yellow arrow in Figure 1a) and the Mg-1.1%Ca alloy grain size of 452 μm. The formation of the $Mg_2Ca$ phase occurs during casting due to the interaction between Mg and Ca elements present in the alloy composition. As the molten Mg-Ca alloy solidifies, the cooling rate and the composition of the alloy play crucial roles in determining the formation and distribution of intermetallic phases [26]. Incorporating MgO particles in the Mg-Ca alloy composite did not have an impact on grain size refinement (448 μm). This is because the MgO particles are incapable of serving as nucleation sites or impeding grain growth during solidification, potentially leading to agglomeration. However, in Figure 1c, the four-pass ECAP Mg-1.1%Ca-2%MgO is depicted. It demonstrates that ECAP can enhance homogeneity and reduce agglomeration (shown in Figure 2c) in the composite by facilitating a more even distribution of the reinforcement phase (MgO) within the magnesium–calcium matrix. This process aids in the dispersion and alignment of the reinforcement particles, ultimately resulting in grain refinement (259 μm) (as shown in Figure 1c). It is worth noting that microstructure refinement achieved through ECAP can significantly enhance the strength and hardness of the Mg-Ca-MgO composite [10].

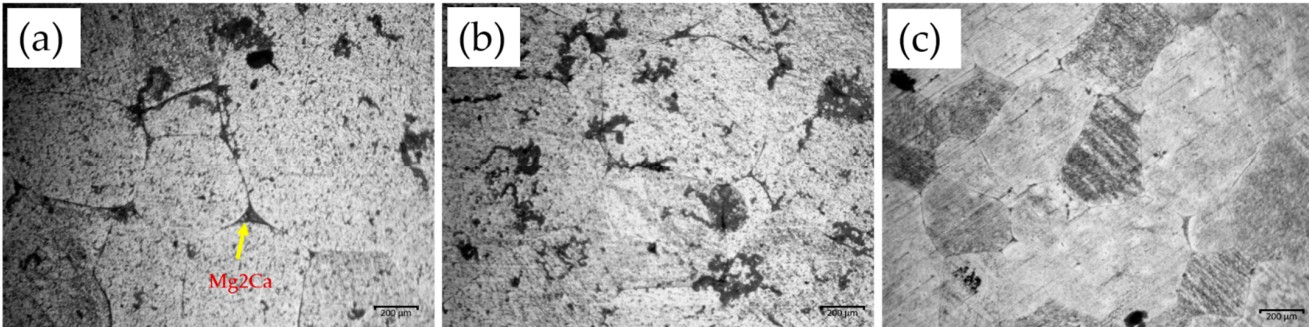

**Figure 1.** Optical microstructure of (**a**) Mg-1.1%Ca, (**b**) Mg-1.1Ca-2%MgO, and (**c**) Mg-1.1%Ca-2%MgO-4Pass.

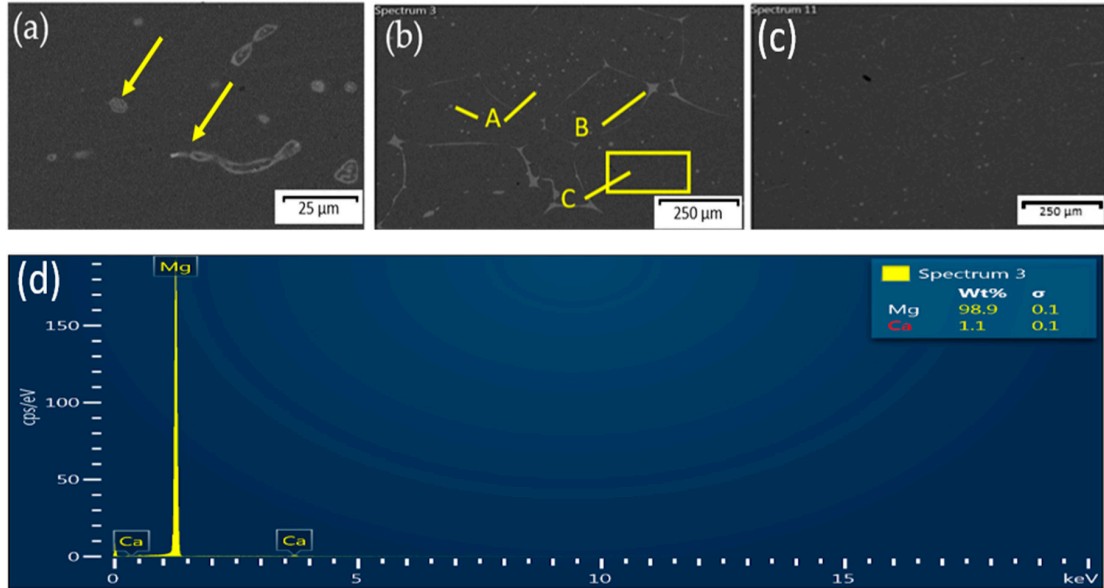

**Figure 2.** SEM images of Mg-1.1%Ca-2%MgO: (**a**) MgO agglomeration; (**b**) high-magnification and (**c**) 4-pass ECAP; (**d**) EDS analysis of Mg-1.1%Ca-2%MgO-4Pass.

The SEM analysis of the Mg-Ca-MgO composites, along with the corresponding EDS results, is presented in Figure 2. Figure 2a demonstrates that the addition of 2% MgO promotes agglomeration in the Mg-Ca alloy, primarily due to inadequate dispersion of the MgO particles within the alloy. Additionally, Figure 2b confirms the presence of Mg (C), MgO (A), and $Mg_2Ca$ (B) elements. However, following the four-pass ECAP process, Figure 2c shows that the Mg-1.1%Ca-2%MgO composite exhibited complete dissolution of the MgO reinforcement particles and a significant reduction in agglomeration, resulting in a relatively uniform distribution of $Mg_2Ca$. This observation is further supported by EDS analysis in Figure 2d. ECAP involves forcing the material through a constrained channel with repeated shearing and deformation. This intense plastic deformation helps break up particle agglomerates and promotes their dispersion within the material. As the material passes through the channel, the particles experience high shear forces that facilitate their separation and redistribution, preventing agglomeration.

Figure 3a shows the XRD pattern analysis of the Mg-Ca alloy composite. It reveals the presence of primary Mg phases with high intensity, as well as some secondary $Mg_2Ca$ phases with low intensity. However, after incorporating 2% MgO and subjecting the Mg-Ca alloy to four passes of ECAP, the secondary phase was dispersed and dissolved within the Mg matrix. Consequently, the intensity of the secondary phase $Mg_2Ca$ was not observed. This observation can be attributed to the dissolution of the secondary phase Mg2Ca through high-temperature plastic deformation during the ECAP process, as depicted in Figure 3b.

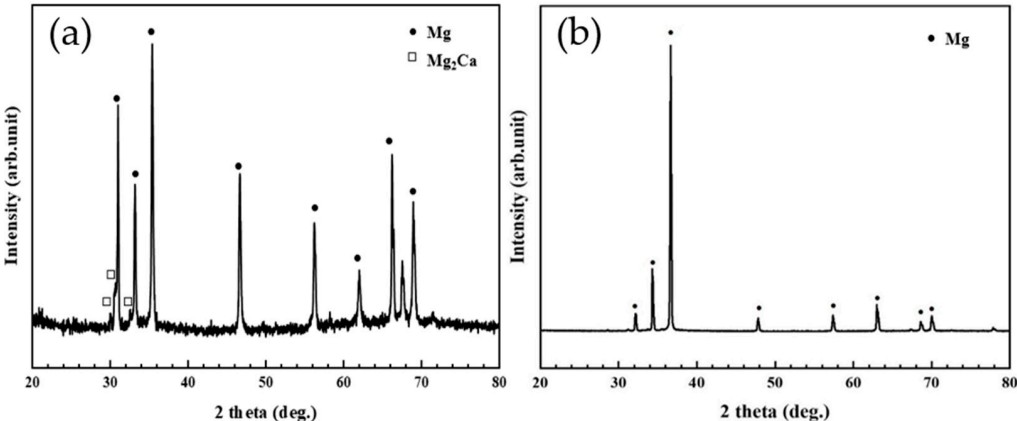

**Figure 3.** XRD analysis of (**a**) Mg-1.1%Ca and (**b**) Mg-1.1%Ca-2%MgO composites after 4-pass ECAP.

*3.2. Mechanical Properties*

3.2.1. Hardness

The hardness values of the Mg-Ca-MgO composites are depicted in Figure 4 and Table 2. It is evident that increasing the amount of Ca enhanced the hardness of the Mg alloy. The highest hardness value of Mg-Ca alloy was 40.7 HV, which was achieved when 1.1% Ca was added to Mg. Moreover, incorporating 2% MgO and subjecting the Mg-Ca alloy to four passes of ECAP significantly increased the hardness compared to Mg-Ca alloys. This is primarily attributed to the presence of hard reinforcement particles (MgO), which restrict highly localized plastic deformation. Additionally, the plastic deformation induced by ECAP leads to strain hardening, where dislocations accumulate and interact, forming complex dislocation structures. These structures create additional barriers to dislocation motion, thereby increasing the material's resistance to plastic deformation and enhancing its hardness. The results demonstrate that the Mg-1.1%Ca-2%MgO composite subjected to four passes of ECAP exhibited the highest hardness value of 47.8 HV (Figure 4). Thus, it can be inferred from Figure 4 that the amount of calcium content, particularly at 1.1%, significantly influenced the hardness. Furthermore, the inclusion of MgO reinforcement and the ECAP process had negligible effects on the hardness value.

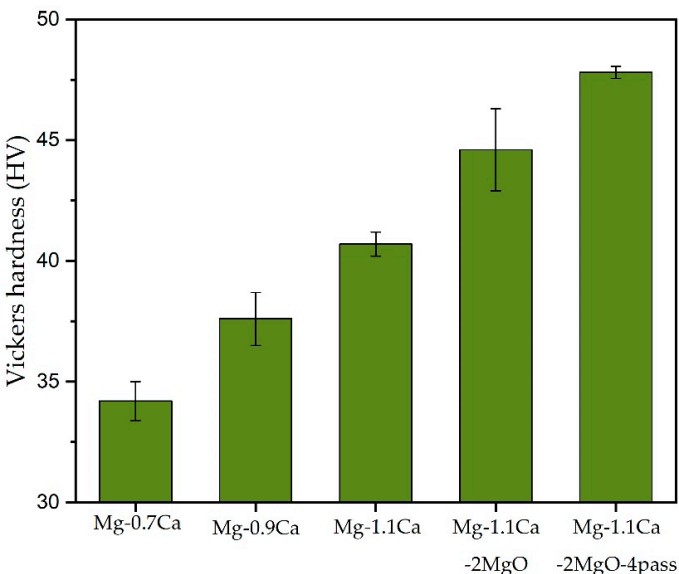

**Figure 4.** The hardness of Mg-Ca alloy composites.

**Table 2.** Tensile properties of the Mg-Ca alloy composite.

| Materials | Tensile Strength | | | Hardness |
|---|---|---|---|---|
| | YTS (MPa) | UTS (MPa) | E (%) | HV |
| Mg-0.7%Ca | 69.8 | 106.8 | 2.9 | 34.2 ± 0.8 |
| Mg-0.9%Ca | 79.3 | 107.6 | 3.2 | 37.6 ± 1.1 |
| Mg-1.1%Ca | 82.6 | 109.2 | 3.3 | 40.7 ± 0.5 |
| Mg-1.1%Ca-2%MgO | 90.3 | 112.7 | 3.7 | 44.6 ± 1.7 |
| Mg-1.1%Ca-2%MgO 4Pass | 124.5 | 161.1 | 4.3 | 47.8 ± 0.25 |

### 3.2.2. Tensile Test

Figure 5 depicts the stress–strain curve of the Mg-Ca alloy composites. It is evident that the inclusion of Ca in Mg enhanced the ultimate tensile strength (UTS), yield tensile strength (YTS), and ductility as the Ca content increased. The values of UTS, YTS, and ductility are presented in Table 2. Among the Mg-Ca alloys, Mg-1.1%Ca exhibited the highest strength and ductility compared to the addition of 0.7% and 0.9% Ca in Mg. Thus, Mg-1.1%Ca was selected for further analysis. Furthermore, MgO reinforcement was introduced into the Mg-1.1%Ca alloy and subjected to four equal channel angular pressing (ECAP) passes. The findings indicate that the inclusion of MgO enhanced the mechanical properties; however, the improvement was not significant, possibly due to the uneven dispersion of MgO particles within the Mg-Ca alloy (Figure 2b). Remarkably, after the fourth ECAP pass on Mg-1.1%Ca-2%MgO, a significant enhancement in mechanical properties was observed compared to the Mg-Ca alloy composites. The four-pass Mg-1.1%Ca-MgO composite exhibited the highest ultimate tensile strength (UTS) of 161.1 MPa, yield tensile strength (YTS) of 124.7 MPa and ductility of 4.3%. These improved mechanical properties can be attributed to the presence of the strengthening reinforcement particles and the plastic deformation process (ECAP).

The results indicate that the addition of MgO nanoparticles alone did not lead to a significant improvement in tensile strength. However, after subjecting the Mg-Ca-MgO alloy to the ECAP process, a notable increase in tensile strength was observed. This enhancement can be attributed to the combined effects of two strengthening mechanisms: particle strengthening and grain refinement. The incorporation of MgO nanoparticles introduces a dispersion strengthening effect, where the fine MgO particles impede dislocation movement and increase the overall strength of the material. Additionally, the ECAP process induces severe plastic deformation, leading to a refined grain structure with

reduced grain size. The refined grains contribute to the Hall–Petch strengthening effect, where smaller grain size inhibits dislocation motion and enhances strength. The synergistic combination of particle strengthening and grain refinement mechanisms results in a significant improvement in tensile strength in the Mg-Ca-MgO alloy after ECAP processing. These findings provide valuable insights into the design and development of high-strength Mg-Ca alloys, highlighting the importance of both MgO addition and ECAP processing as effective strategies for enhancing tensile strength in magnesium-based materials [19,20].

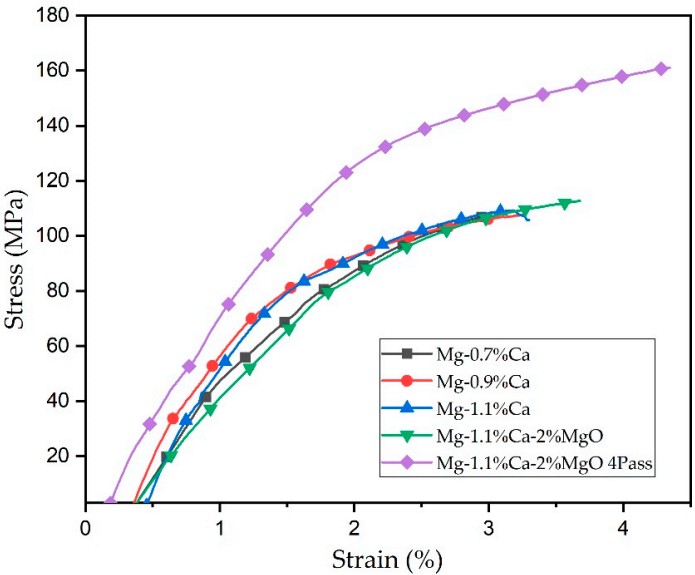

**Figure 5.** Stress–strain curve of Mg-Ca alloy composite.

## 4. Electrochemical Corrosion Test

The electrochemical performance was measured to evaluate the corrosion properties of the Mg-Ca-MgO composites. Figure 6 illustrates the dynamic potential polarization curve of Mg with varying Ca percentages. The results indicate that there was no significant change in the corrosion current density. Specifically, the corrosion current density of the Mg-Ca alloy was 43.2 $\mu A/cm^2$. The limited solubility of calcium in magnesium restricts the interaction between calcium and magnesium ions, resulting in minimal impact on the overall electrochemical behavior. Despite different percentages of calcium added to the magnesium alloy, the actual concentration of calcium in the magnesium matrix may not have been substantial enough to induce noticeable changes in the corrosion current [27].

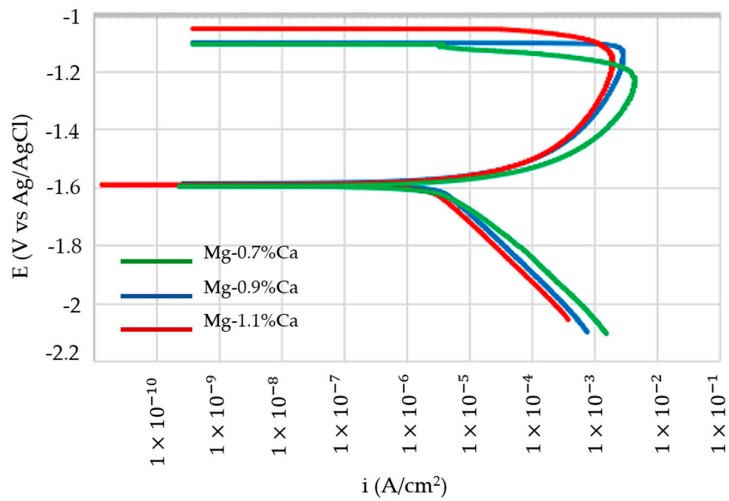

**Figure 6.** Dynamic potential polarization curve of Mg-0.7%Ca, Mg-0.9%Ca, and Mg-1.1%Ca.

Figure 7 displays the potential polarization curves of Mg-1.1%Ca, considering the effect of MgO and the ECAP process. It can be observed that the incorporation of MgO into the Mg-Ca composite led to a decrease in corrosion current. However, the reduction in corrosion current density after the addition of MgO was not significant. Furthermore, the ECAP process effectively reduced the corrosion current density by 10.1 $\mu$A/cm$^2$ in the Mg-1.1%Ca-2%MgO composite. Decreasing the corrosion current can enhance the corrosion resistance of a material. The corrosion current serves as a measure of the rate at which corrosion reactions occur on the material's surface. By reducing the corrosion current, the corrosion process is effectively slowed down, thereby increasing the material's resistance to corrosion.

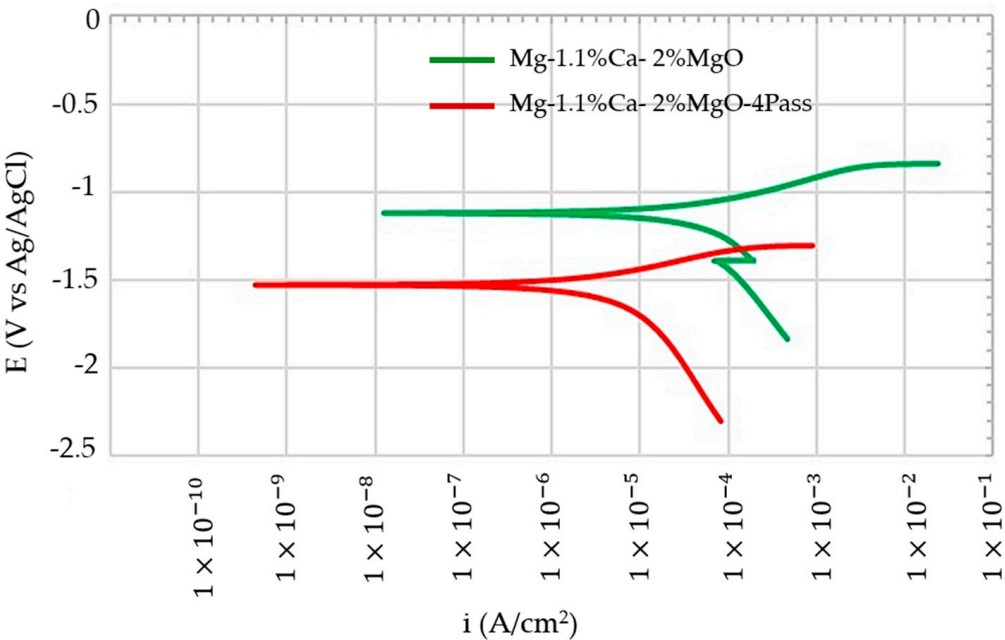

**Figure 7.** Dynamic potential polarization curve of Mg-1.1%Ca-MgO with different ECAP passes.

Based on these findings, it can be concluded that the corrosion resistance of the Mg-1.1%Ca-MgO composite was indeed enhanced due to the influence of the ECAP process. This improvement in corrosion resistance can be attributed to the uniform distribution and grain refinement resulting from the ECAP process. The plastic deformation induced by ECAP introduces strain hardening in the material, leading to increased hardness and strength. Strain-hardened materials generally exhibit improved resistance to corrosion due to their enhanced ability to resist deformation and localized corrosion attacks. In addition, the agglomerates generally have a larger size compared to individual dispersed particles. Breaking up agglomerates into smaller particles during ECAP increases the overall surface area available for interaction with the corrosive environment. The increased surface area allows for more active sites to participate in corrosion processes, leading to more efficient passivation of the material's surface. This increased surface area can enhance the formation and stability of protective oxide layers, reducing the susceptibility to corrosion [28].

## 5. Biocompatibility Analysis

### 5.1. Cytotoxicity Test

The osteoblast cell line MC3T3-E1 derived from the mouse skull was utilized in this study to investigate cytotoxicity and cell apoptosis. The impact of different calcium (Ca) contents on the cell survival rate of magnesium (Mg) was examined, and the results are presented in Figure 8. It is evident that at Ca contents of 0.7% and 0.9%, the cell survival rates after 24 h of culture were 83.95% and 84.3%, respectively, with no significant difference between them. After 48 h, the survival rates remained nearly the same for both 0.7% and

0.9% Ca. However, when the calcium content increased to 1.1%, there was a significant increase in cell survival rates after 24 and 48 h, reaching 90.6% and 92.67%, respectively. These findings indicate a positive effect on cell performance when the calcium content reaches 1.1% [10].

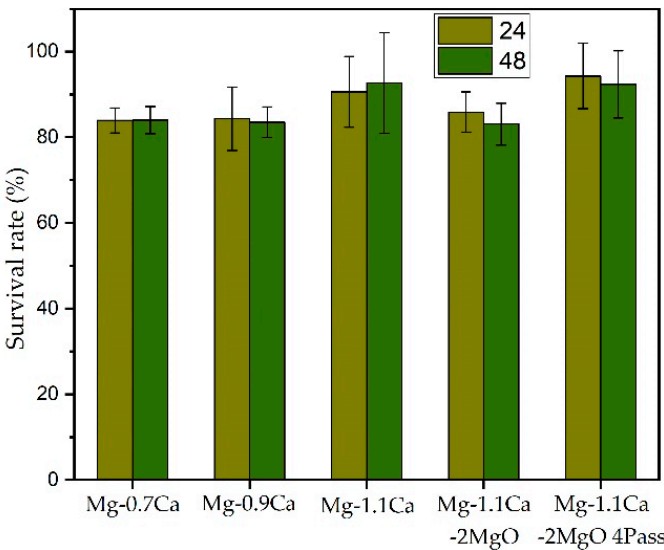

**Figure 8.** Cell survival performance of Mg-Ca-MgO composites after 24 and 48 h.

Furthermore, the addition of MgO to the Mg-1.1%Ca alloy resulted in a decrease in cell survival rate. The agglomeration of MgO particles created localized areas with high particle concentrations, which directly affected the cells in contact with them, potentially leading to reduced cell survival. Conversely, the application of equal channel angular pressing (ECAP) on the Mg-1.1%Ca-2%MgO composite significantly enhanced cell performance. The results reveal that the cell survival rate increased with each pass of ECAP, both at 24 and 48 h of treatment. Notably, the highest cell survival rate (94.3%) was achieved with a four-pass ECAP on the Mg-1.1%Ca-2%MgO composite at the 24 h mark. This improvement in cell viability can be attributed to the grain refinement occurring during the ECAP process. ECAP refines the material's surface, resulting in a more uniform and smoother surface topography, which enhances biocompatibility and reduces the risk of cell adhesion or tissue irritation, thereby promoting cell survival [28,29].

### 5.2. Cell Apoptosis

The magnesium–calcium alloy composites underwent a cell apoptosis test, and the results are presented in Figure 9. After 24 h of culture, the cell apoptosis rates for Mg-0.7%Ca and Mg-0.9%Ca were −20.6% and −26.9%, respectively. After 48 h of culture, the performances of Mg-0.7%Ca and Mg-0.9%Ca were nearly identical. However, with an increase in calcium content to 1.1%, the 24 h apoptosis rate of cells significantly decreased to −39.7%, and after 48 h, it further decreased to −42.85%. Clearly, when the calcium content reaches 1.1%, it exerts a positive effect on cell apoptosis. Furthermore, the introduction of 2% MgO in the Mg-1.1%Ca alloy resulted in a decrease in cell apoptosis performance. However, after the four-pass ECAP process, the cell apoptosis of Mg-1.1%Ca-2%MgO significantly decreased. The cell apoptosis rate reached its peak (−49.8%) at 24 h, and after 48 h, it remained high at −35.8%. This confirms that increasing the number of ECAP passes for grain refinement can inhibit the occurrence of apoptosis.

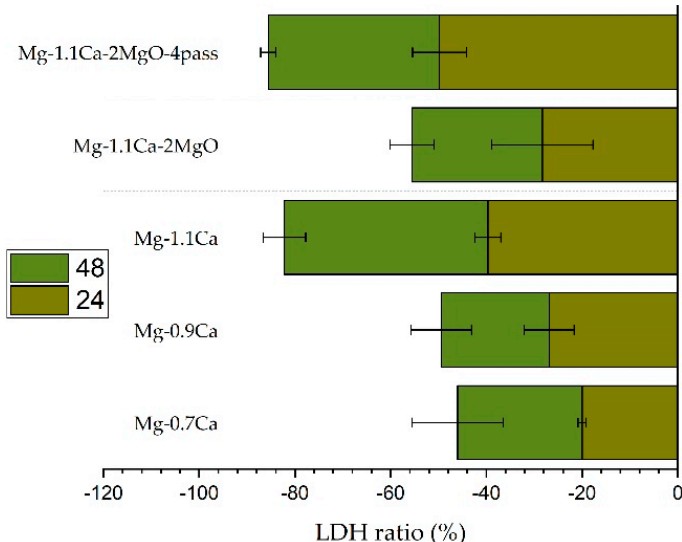

**Figure 9.** Apoptosis performance of Mg-1.1%Ca-MgO alloy composites after 24 and 48 h.

## 6. Conclusions

The Mg-Ca alloy composites were fabricated using a stir-casting process. The effects of MgO and the ECAP process on microstructure, corrosion resistance, mechanical properties, and biocompatibility have been studied. The conclusions of the results are summarized below:

1. The incorporation of MgO into magnesium–calcium alloy composites did not yield significant results in terms of grain refinement, improvement of tensile strength, corrosion rate reduction, and enhanced biocompatibility. However, after subjecting the Mg-Ca-MgO alloy to the ECAP process, significant improvements were observed in all these aspects.

2. The ECAP process, which involves severe plastic deformation, resulted in notable grain refinement, leading to smaller grain sizes within the alloy. This refinement likely contributed to the enhanced mechanical properties. The maximum YTS (124.5 MPa), UTS (161.1 MPa), and elongation (4.3%) were obtained by the four-pass Mg-1.1%Ca-2%MgO composite.

3. Additionally, the ECAP-treated alloy exhibited a decrease in corrosion rate, indicating improved corrosion resistance. The corrosion current decreased to 43.2 $\mu A/cm^2$ and 10.1 $\mu A/cm^2$ with the ECAP passes on the Mg-1.1%Ca-2%MgO composite, which clearly shows that the corrosion resistance improved.

4. Moreover, the biocompatibility of the composite also improved after the ECAP process. The survival rate after 24 h of cell incubation reached a maximum of 94.3% after 4 passes of ECAP of Mg-1.1%Ca-2%MgO, confirming that the improvement in grain refinement based on the number of ECAP passes improved the performance of the cell survival rate.

5. The Mg-1.1%Ca-2%MgO composite showed a significant decrease in apoptotic performance after four ECAP cycles, with the highest apoptotic rate (−49.8%) at 24 h and −35.8% at 48 h, confirming that the improvement in grain refinement following ECAP prevented apoptotic performance.

Overall, the incorporation of MgO into the magnesium–calcium alloy composites alone did not yield the desired outcomes. However, when combined with the subsequent ECAP process, significant positive effects were observed, including grain refinement, increased mechanical strength, reduced corrosion rate, and improved biocompatibility.

**Author Contributions:** Conception, F.-Y.F., S.-J.H. and C.-F.W.; methodology, F.-Y.F., S.-J.H. and C.-F.W.; formal analysis, M.S. and C.-F.W.; investigation, S.-J.H. and F.-Y.F.; writing—original manuscript preparation, C.-F.W. and M.S.; writing—review and editing, S.-J.H. and M.S.; supervision, S.-J.H. and F.-Y.F. All authors have read and agreed to the published version of the manuscript.

**Funding:** Taipei Medical University—National Taiwan University of Science and Technology Joint Project no: TMU-NTUST-111-09.

**Institutional Review Board Statement:** Not applicable.

**Informed Consent Statement:** Not applicable.

**Data Availability Statement:** The data presented in this study are available on request from the corresponding author.

**Acknowledgments:** Authors would like to thank the National Taiwan University of Science and Technology and Taipei Medical University's Theme Project TMU-NTUST-111-09 for the financial support in this work.

**Conflicts of Interest:** The authors declare no conflict of interest.

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
