# Peer review of "Effect of ECAP on Microstructure, Mechanical Properties, Corrosion Behavior, and Biocompatibility of Mg-Ca Alloy Composite"

_jcs, doi:10.3390/jcs7070292_

Round 1
Reviewer 1 Report
Dear Authors,
the article is written in a simple and clear way.
However, I ask for a deeper evaluation of the results. In the Conclusions section, it is stated that composition has an effect on the parameters studied. This is obvious, otherwise you would not have investigated it.
Please describe this in more detail. What happens inside the alloy? Does its structure change? In all subsections, add an explanation of the results. Your comments are very brief and need to be completed in more detail.
Additional comments:
Lines 98, 99 - give the composition of the Mg-Ca alloy you used. Did you prepare it yourself or was it commercial?
Lines 126, 127 - you list the Mg2Ca alloy. Confirm that this type of alloy was actually created.
Author Response
Response to Reviewer 1 Comments
Dear reviewer,
Thank you very much for your valuable remarks. The manuscript was strictly revised according to your comments.
Please find detailed comments to your remarks:
The article is written in a simple and clear way.
However, I ask for a deeper evaluation of the results. In the Conclusions section, it is stated that composition has an effect on the parameters studied. This is obvious, otherwise, you would not have investigated it. Please describe this in more detail. What happens inside the alloy? Does its structure change? In all subsections, add an explanation of the results. Your comments are very brief and need to be completed in more detail.
ANSWER: We thank you the reviewer for the instructive comment; we included all the information in the manuscript based on the comments. (Page 12, Line 328-354)
Additional comments:
Lines 98, 99 - give the composition of the Mg-Ca alloy you used. Did you prepare it yourself or was it commercial?
ANSWER: We thank you the reviewer for the valuable comments and we have included all the information in the manuscript based on your valuable comments.
The magnesium calcium (Mg-Ca) alloy, obtained from Suzhou Haichuan Rare Metal Products Co., Ltd., was used as the matrix material with varying calcium content ratios (Ca=0.7%, 0.9%, 1.1%). To reinforce the alloy, 2% of MgO particles with a particle size of 30 nm were incorporated, purchased from Diyi Chemical Materials Co., Ltd. (Page 3, Line 102-105)
Lines 126, 127 - you list the Mg2Ca alloy. Confirm that this type of alloy was actually created.
ANSWER: It is very important point. Figure 1a illustrates the optical microstructure (OM) of Mg-1.1%Ca alloy composites. It can disclose the presence of Mg2Ca intermetallic phases at the grain boundary (indicated by the yellow arrow in Figure 1a). The formation of the Mg2Ca phase occurs during casting due to the interaction between Mg and Ca elements present in the alloy composition. As the molten Mg-Ca alloy solidifies, the cooling rate and the composition of the alloy play crucial roles in determining the formation and distribution of intermetallic phases [27]. (Page 4, Line 144-150)
Reviewer 2 Report
Assessing the reviewed manuscript I might say that it is written in bad English and must be obligatory prepared again with an aid of a person fluent in this language. Afterward, it should be sent to reviewers. There are many unclear phrases and grammar errors. Not specifically identifying the errors, I have indicated many of them by colour. Besides, I have included in the text several of my remarks, also by color, which are discussed below.
Despite that being a serious complication for any reader, I am trying to prepare a preliminary opinion on the reviewed manuscript. In the beginning, I positively assess the research plan. The investigated material is based on a magnesium-calcium matrix reinforced by magnesium oxide nanoparticles. The input variables included calcium content in three of its values, and cold extrusion with passes from 1 to 4, then a matrix composed of 12 combinations. The tests applied to characterize microstructure and properties included light microscopy, SEM, EDS, XRD, hardness and tensile tests, corrosion tests, and two biocompatibility tests. If all these test results might be presented and properly discussed, and written in perfect language, the manuscript might be accepted by me. However, I have a lot of critical remarks proposing the necessary modifications to the manuscript.
1. The title does not correspond to the contents. It suggests that the investigations have focused on grain refinement, which is not true (see below), and biocompatibility. I suggest changing the title making it clear that the research was on the effects of Ca content and ECAP deformation technique on mechanical and corrosion behavior, and biocompatibility. The title might be shorter, but not the present one.
2. In Materials and Methods, there is a lack of several data such as:
- Description of mechanical pretreatment (sandpapers, till what final grade? Or diamond paste?)
- Types, manufacturers, and country for each equipment used, e.g. oven, ECAP, etc.
- Name of the corrosion test (potentiodynamic method?) and its detailed description.
3. The discussion of possible effects of Ca content on mechanical, corrosion, and biological behavior should be more developed. What do you mean by the grain refinement in this case, what is evidence about that? What means “nucleation sites for grain growth restriction”? “The addition of 2%MgO and the 4 Pass 131 ECAP process effectively reduced the agglomeration, and secondary phase formation and 132 refine the grain boundaries”; an agglomeration of what is prevented by magnesium oxide phases, and how ECAP prevents the agglomeration, of what? “distribution of 147 Mg2Ca is relatively uniform “, what does it mean? There is a plenty of such unclarities and improper and no proved statements in the following text of this section.
4. The discussion of the possible effects of ECAP (number of passes) on mechanical, corrosion, and biological behavior should be more developed. For example, the “ECAP process was used to increase the corrosion resistance to address the disadvantages used in biomaterials, which generate shear forces to break the agglomeration behavior of magnesium oxide particles”; how the breaking the agglomerates affect the corrosion resistance?
5. “The uniform distribution of the particles can impede the dislocation motion”; explain it on the model as I presume that the dislocation movement depends on the number and size of particles.
There are only some examples.

I marked many of errors with color in the attached file.
Author Response
Response to Reviewer 2 Comments
Dear reviewer,
Thank you very much for your valuable remarks. The manuscript was strictly revised according to your review comments.
Please find detailed comments to your remarks:
Assessing the reviewed manuscript I might say that it is written in bad English and must be obligatory prepared again with an aid of a person fluent in this language. Afterward, it should be sent to reviewers. There are many unclear phrases and grammar errors. Not specifically identifying the errors, I have indicated many of them by colour. Besides, I have included in the text several of my remarks, also by color, which are discussed below.
Despite that being a serious complication for any reader, I am trying to prepare a preliminary opinion on the reviewed manuscript. In the beginning, I positively assess the research plan. The investigated material is based on a magnesium-calcium matrix reinforced by magnesium oxide nanoparticles. The input variables included calcium content in three of its values, and cold extrusion with passes from 1 to 4, then a matrix composed of 12 combinations. The tests applied to characterize microstructure and properties included light microscopy, SEM, EDS, XRD, hardness and tensile tests, corrosion tests, and two biocompatibility tests. If all these test results might be presented and properly discussed, and written in perfect language, the manuscript might be accepted by me. However, I have a lot of critical remarks proposing the necessary modifications to the manuscript.
ANSWER: It is very important comment, we modified the writings accordingly.
- The title does not correspond to the contents. It suggests that the investigations have focused on grain refinement, which is not true (see below), and biocompatibility. I suggest changing the title making it clear that the research was on the effects of Ca content and ECAP deformation technique on mechanical and corrosion behavior, and biocompatibility. The title might be shorter, but not the present one.
ANSWER: It is a valuable comment and we have changed our topic in the manuscript based on the comments.
Title: Effect of ECAP on microstructure, mechanical properties, corrosion behavior, and biocompatibility of Mg-Ca-MgO composites. (Page 1, Line 2-3)
- In Materials and Methods, there is a lack of several data such as:
Description of mechanical pretreatment (sandpapers, till what final grade? Or diamond paste?) Types, manufacturers, and country for each equipment used, e.g. oven, ECAP, etc.
ANSWER: We thank you the reviewer for the valuable comments and we have included all the information in the manuscript based on your valuable comments.
To prepare the samples for microstructural analysis, they were ground using silicon carbide abrasive papers (#240, #400, #600, #1000, #2000), followed by polishing with aluminum oxide paper. Microstructural analysis was performed using an optical microscope (OM-Zeiss Axiotech 25HD, Germany), scanning electron microscope (SEM-JSM-7900F, Japan), and X-ray diffraction (XRD-Rigaku XtaLAB Synergy DW, Japan). For ECAP analyses, the ingots were cut into dimensions of 11.5 mm × 11.5 mm × 75 mm. Heat treatment was conducted in a furnace at 410°C for 24 hours. Following the heat treatment, ECAP processing was performed with the following parameters: channel angle Φ=120°, outer rounding angle Ψ=60°, working temperature of 350°C, Bc route, and extrusion at 4 passes. (Page 3, Line 107-116)
- Name of the corrosion test (potentiodynamic method?) and its detailed description.
Corrosion analysis was carried out using a dynamic potential polarization curve test (using the PalmSens4C electrochemical analyzer/impedance analyzer). In this experiment, a three-electrode system was employed for measurement. The sample served as the working electrode, the platinum sheet was used as the counter electrode, and the saturated silver chloride electrode (Ag/AgCl sat. KCl) served as the reference electrode. Hank's Balanced Salt Solution (HBSS) was used as the solution, and its composition is provided in Table 1. The exposed area of the sample was 1 cm2, with a sample area to solution volume ratio of 1:250. The solution temperature was maintained at 37±0.5℃. Prior to measurement, it was necessary to wait for the open circuit potential (OCP) between the sample and the electrolyte to reach dynamic equilibrium, which typically took approximately 15 minutes. Subsequently, the dynamic potential measurement was performed and the data was recorded. The experimental scan voltage range was set at ±1V, with a scanning rate of 1mV/s. (Page 4, Line 121-135)
Table 1. The chemical composition of Hank's Balanced Salt Solution [26].
Chemicals |
g/L |
NaCl |
8.0 |
KCl |
0.4 |
CaCl2 |
0.14 |
MgSO4·7H2O |
0.06 |
MgCl2·6H2O |
0.1 |
Na2HPO4·12H2O |
0.06 |
KH2PO4 |
0.06 |
C6H12O6 |
1.0 |
NaHCO3 |
0.35 |
- The discussion of possible effects of Ca content on mechanical, corrosion, and biological behavior should be more developed. What do you mean by the grain refinement in this case, what is evidence about that? What means “nucleation sites for grain growth restriction”? “The addition of 2%MgO and the 4 Pass 131 ECAP process effectively reduced the agglomeration, and secondary phase formation and 132 refine the grain boundaries”; an agglomeration of what is prevented by magnesium oxide phases, and how ECAP prevents the agglomeration, of what? “Distribution of 147 Mg2Ca is relatively uniform “, what does it mean? There is a plenty of such unclarities and improper and no proved statements in the following text of this section.
ANSWER: It is a valuable comment and we included all the information in the manuscript based on the comments.
The grain size measurement of the Mg-Ca alloy composites has been discussed in the microstructural analysis section. The presence nucleation sites for grain growth restriction can hinder or impede this coarsening process. These nucleation sites act as barriers or obstacles to the growth of neighboring grains by providing localized regions of high energy or strain, making it difficult for new grains to form or grow. However, in this work, incorporating MgO particles in the Mg-Ca alloy composite does not have an impact on grain size refinement. This is because the MgO particles are incapable of serving as nucleation sites or impeding grain growth during solidification. (Page 4-5, Line 144-160)
ECAP involves forcing the material through a constrained channel with repeated shearing and deformation. This intense plastic deformation helps break up particle agglomerates and promotes their dispersion within the material. As the material passes through the channel, the particles experience high shear forces that facilitate their separation and redistribution, preventing agglomeration. (Page 5, Line 172-177)
- The discussion of the possible effects of ECAP (number of passes) on mechanical, corrosion, and biological behavior should be more developed. For example, the “ECAP process was used to increase the corrosion resistance to address the disadvantages used in biomaterials, which generate shear forces to break the agglomeration behavior of magnesium oxide particles”; how the breaking the agglomerates affect the corrosion resistance?
ANSWER:
As pointed out by the reviewer, the agglomerates generally have a larger size compared to individual dispersed particles. Breaking up agglomerates into smaller particles during ECAP increases the overall surface area available for interaction with the corrosive environment. The increased surface area allows for more active sites to participate in corrosion processes, leading to a more efficient passivation of the material's surface. This increased surface area can enhance the formation and stability of protective oxide layers, reducing the susceptibility to corrosion. (Page 9, Line 274-280)
- “The uniform distribution of the particles can impede the dislocation motion”; explain it on the model as I presume that the dislocation movement depends on the number and size of particles.
ANSWER: We thank you the reviewer for the instructive comment.
In a material, dislocations are line defects that occur due to the movement of atoms within crystal structures. Dislocations play a significant role in the plastic deformation of materials. When particles are uniformly distributed within a material, they can impede the motion of dislocations.

Round 2
Reviewer 1 Report
Dear Authors,
the manuscript has been edited according to my requirements.
It would be useful to complete the conclusions section, suggest other ways of using the material and ideas for further experiments in this area.
Reviewer 2 Report
I accept the article in its current form.